# Does disability modify the association between poor mental health and violence victimisation over adolescence? Evidence from the CoVAC cohort study in Uganda

Daniel J. Carter[1]*, Charlie F. M. Pitcairn[1], Emily Eldred[1], Louise Knight[1], Janet Nakuti[2], Angel Mirembe[2], Lydia Atuhaire[3], Elizabeth Allen[4], Amiya Bhatia[5], Dipak Naker[2], Jenny Parkes[6], Karen Devries[1]

1 Department of Population Health, London School of Hygiene and Tropical Medicine, London, United Kingdom, 2 Raising Voices, Kampala, Uganda, 3 Medical Research Council/Uganda Virus Research Institute, Entebbe, Uganda, 4 Department of Medical Statistics, London School of Hygiene and Tropical Medicine, London, United Kingdom, 5 Department of Social Policy and Intervention, Oxford University, Oxford, United Kingdom, 6 Institute of Education, University College London, London, United Kingdom

* daniel.carter1@lshtm.ac.uk

**Data Availability Statement:** Data access is restricted by LSHTM for safeguarding of sensitive information about people under the age of 18.

## Abstract

We aimed to estimate the impact of poor mental health in early adolescence on subsequent poor mental health, depression, and violence victimisation in late adolescence and to determine whether young people living with disabilities experienced a stronger relationship between mental health and these outcomes. Data from two waves of a longitudinal cohort study of 2773 Ugandan adolescents were used to assess the impact of mental health difficulties in early adolescence (aged 11–14) on presence of subsequent mental health difficulties, depression and past year violence victimisation in later adolescence (aged 15–18). We used g-computation to examine how these outcomes changed dependent on levels of poor mental health in early adolescence and explored functional difficulties as an effect modifier. This study demonstrates high prevalence of mental health difficulties in adolescence. There is a positive association between mental health difficulties in early adolescence and experience of mental health difficulties, depression, and past year violence in later adolescence. The risk of poor outcomes is greater for individuals experiencing poorer mental health in early adolescence. The relationships between early mental health difficulties and both mental health difficulties in later adolescence and past year violence are stronger in young people with functional difficulties. Poor mental health in early adolescence is associated with depression and violence victimisation in later adolescence, and the association is stronger among adolescents living with disabilities. School-aged adolescents would benefit from violence prevention and mental health promotion interventions which are inclusive and engage and respond to the needs and rights of adolescents with disabilities.

However, data is available from the Research Data Management team at researchdatamanagement@lshtm.ac.uk on reasonable request to eligible parties and with the completion of required prerequisites. Data requests can also be dispatched to this manuscript's authors.

**Funding:** This work was funded by Medical Research Council ref. MR/R002827/1 awarded to KD. The funders had no role in study design, data collection and analysis, decision to publish, or preparation of the manuscript.

**Competing interests:** The authors have declared that no competing interests exist.

## Introduction

Adolescence is a life stage in which poor mental health most frequently occurs for the first time [1]. Longitudinal studies in high-income countries have found that signs of poor mental health in adolescence are associated with subsequent experience of depression and other mental health disorders, as well as increased risk of later violence victimisation [2–5]. Poor mental health in early adolescence is associated with a number of adverse health outcomes in later adolescence, including experiences of violence victimisation, but the directionality of this relationship is often unclear [6].

Regardless of direction, the risk of poor mental health and violence is not equal across groups—both are substantially more common among adolescents with disabilities or functional difficulties than without. People living with disabilities make up 16% of the world's population, a higher proportion of whom live across the Global South, and 1 in 10 children worldwide have a disability [7–9]. Disability exists across a spectrum, and recent estimates across 41 countries found 12.6% of people report some level of functional difficulty, with walking and seeing difficulties reported most frequently [10]. People living with disabilities are more likely to experience poor mental health, including anxiety and psychological distress, compared to their peers without a disability [11]. Further, amongst children and adolescents with multiple functional difficulties, a large proportion report signs of anxiety and depression. For instance, the prevalence of these signs is 32% in children and adolescents with communication difficulties, 28% in young people with self-care difficulties, and 27% in young people [7]. Worldwide, up to a quarter of people living with disabilities will experience violence in their lifetime and individual studies have shown that girls with functional difficulties have twice the odds of sexual violence versus non-disabled peers [12, 13].

At present, it is unclear whether adolescents with disabilities simply experience a higher prevalence of both poor mental health and violence, or whether disability status is an effect modifier of the association between poor mental health and violence. Some studies in high income settings have found that people living with disabilities are at higher risk of negative mental health consequences resulting from violence than people without disabilities, but these studies have not specifically looked at adolescents [14, 15]. Understanding the experiences and/or trajectories for adolescents with and without disabilities are essential to inform equity-oriented, disability justice-based interventions to reduce disease burden.

In Uganda specifically, the prevalence of depression and anxiety among children and adolescents is estimated to be about 23.6%, with estimates varying between 2.9% and 46.0% [16]. According to the Ugandan Violence Against Children Survey, rates of childhood emotional, physical, and sexual violence are about 34%, 59%, and 35% respectively in girls, and about 36%, 68%, and 17% in boys [17]. Existing research on links between adolescent mental health and violence in Uganda has primarily focused on analysing violence as a predictor of subsequent mental health outcomes or has investigated associations in specific adolescent subgroups such as refugees, people living with HIV, or those affected by conflict [6, 18–20]. Where research has focused on adolescents as a specific population of interest, it has been cross-sectional and thus has not established any directionality or impact of poor mental health across the lifecourse [21]. We found no studies to date that have investigated the longitudinal relationship between poor mental health in early adolescence and subsequent poor mental health and violence victimization in adolescence in this setting.

In this paper, we will use cohort data from adolescents in Uganda. We aim to:

a. Describe the prevalence of mental health difficulties, depression, and violence in early and late adolescence;

b. quantify the impact of poor mental health in early adolescence on risk of subsequent poor mental health, depression, and violence victimisation in late adolescence and;

c. determine whether the relationship between poor mental health in early adolescence and outcomes in later adolescence varies dependent on experiencing functional difficulties in early adolescence.

## Methods

This paper is a secondary analysis of the first two waves of data from the Contexts of Violence in Adolescence Cohort (CoVAC) study. CoVAC is a prospective cohort study with three waves of data collection between early 2014 and late 2022 in Luwero, Uganda (see protocol for further details) [22]. Participants included young people from 42 primary schools in Luwero who were first interviewed in 2014 aged 11 to 14 years and interviewed again in 2018 aged 15–18 years. 3431 participants participated in Wave 1 and 2773 in Wave 2 (80.8% follow up). Data were collected through in-person interviews by trained data collectors. Young people were interviewed in a private place where they could not be overheard at home, in school, or in their community and the interviews included numerous stopping points to confirm young people were still in a comfortable place to continue. Researchers who had received extensive training on how to interview young people about violence conducted all interviews in either Luganda or English using a tablet to record answers. Further details can be found in the CoVAC protocol [22].

Voluntary, informed consent for participation was obtained each time participants were contacted to take part in the study. Consent was obtained in written form from all participants aged over 18 and emancipated minors. Caregivers of participants under 18 years and under were able to opt children out of participation, and participants provided informed assent No participation incentives were given but participants were given a small in-kind reimbursement for time. Protection of the safety of participants was ensured through a planned referral pathway to healthcare and/or social services when interviewers identified a risk to the adolescent's welfare, including a rapid referral pathway.

### Ethics statement

Ethical approval for the CoVAC study was received from the London School of Hygiene and Tropical Medicine (LSHTM) Ethics Committee (6183 and 14768), University of London (UCL)–Institute of Education (IoE) Research Ethics Committee (1091), and the Uganda Virus Research Institute (UVRI) and (UNCST) Ethics committees (SS2520 and SS4722). Ethical approval to conduct the present analysis was sought and obtained from the LSHTM MSc Research Ethics Committee (26845).

**Measures.** *Exposure*. All items from the Strengths and Difficulties Questionnaire (SDQ-25) were used as a measure of adolescent mental health at Wave 1 [23]. The SDQ has been validated in diverse settings and translated into several languages [23, 24]. We generated a binary and continuous SDQ variable. For the binary variable, used in descriptive analyses, we dichotomised the difficulties score by grouping young people in the highest quintile of the distribution of scores into the "higher difficulties" category. The remaining participants, in any of the other SDQ quintiles were then classified as having "lower" difficulties. This method has been previously used in the analysis of mental health in community samples of adolescents, including in Luwero district [23, 25]. For our main analysis, we operationalised the SDQ-25 as a continuous exposure variable to understand differences in outcomes at a range of different SDQ-25 values, as in general, one-unit increases in SDQ scores correspond to meaningful changes

in mental health at the group level [26]. The SDQ score ranges from 0–40. The cut-off point for high scores is context dependent but usually sits between 16–18 [23].

*Outcomes.* The primary outcomes considered in this analysis were all measured at the Wave 2 survey. Mental health difficulties were defined at Wave 2 through the dichotomised SDQ-25 score as described above. The presence of depression was defined using the Patient Health Questionnaire for Adolescents (PHQ-A) depression screening tool, an adolescent-specific version of the PHQ-9, a well-validated and widely used depression screening instrument [27]. In this study, a score of 10 or more was considered diagnostic of depression, and such diagnostic criteria have been previously validated in Uganda in both rural and urban populations [28–30].

Violence victimisation at Wave 2 was defined as the experience of physical, emotional, or sexual violence perpetrated by any of parents, intimate partners, teachers, peers, or individuals that respondents worked with. Experience of peer, school-based, caregiver, and employer violence was measured using questions adapted from the International Society for the Prevention of Child Abuse and Neglect-Child Abuse Screening Tool- Child Institutional version (ICAST-CI). ICAST-CI itself contains 36 questions addressing five dimensions: exposure to violence, sexual abuse physical abuse, neglect, and emotional abuse [31]. Experience of intimate partner violence was measured using questions adapted from the Conflict in Adolescent Dating Relationships Inventory (CADRI) and the WHO Multi-Country study on women's health and domestic violence against women [32, 33].

*Effect modifier.* The Washington Group Short Set questions were used to generate a binary measurement of any functional difficulties, which ask about vision, hearing, communication, mobility, self-care, and cognition. This set of questions has been validated in the Ugandan context [34]. Individuals reporting that they had 'some' difficulty, 'a lot' of difficulty, or 'could not do at all' were coded as having any functional difficulties, versus those with 'no difficulty', and thus this definition includes both young people living with disabilities and young people with functional difficulties.

*Confounders.* A directed acyclic graph (DAG) was used to inform the selection of potential confounders. Covariates were selected for inclusion in a DAG if they were plausibly causes of both the exposure and outcomes. The confounders selected for analysis were respondent age at Wave 1, gender at Wave 1, socioeconomic position (SEP) at Wave 1 and Wave 2, disability or functional difficulty at Wave 1 (where not treated as an effect modifier), and having received an in-school violence prevention intervention (the Good Schools Toolkit; details in Devries et al. 2015 [35]) at Wave 1 [12, 36]. Violence victimisation at Wave 1 was not included as a confounder given the temporal ambiguity of onset of mental health symptoms and reported experiences of violence. Respondent age was categorised into three levels (10 years-old or less, 11–14 years-old, 15 years-old or more) while functional difficulty, GST intervention exposure and gender were treated as binary variables. SEP was accounted for at both Wave 1 (using number of meals per day received by respondents as a proxy) and Wave 2 (using an asset cluster variable) as separate variables. Asset cluster was not measured at Wave 1 as respondents were deemed too young to accurately report these for their households.

**Analysis.** All analyses were carried out in R v4.2.2. Given the relative sparsity of missing data, a complete case analysis was conducted for all outcomes. We fit a logistic regression model for each outcome variable, adjusting for relevant confounders, and applied the g-computation algorithm to obtain marginal estimates of the risk difference and risk ratio across all values of SDQ score at Wave 1 [37]. The associated 95% confidence intervals were obtained using the delta method. Marginal estimates are 'population average' estimates and may be interpreted as the average additional risk of experiencing the modelled outcome for a one unit increase in SDQ score. We present both overall and stratified estimates by presence of any

functional difficulties where the statistical interaction term of the adjusted model suggests effect modification based on a Wald test.

## Results

### Participant characteristics and prevalence of poor mental health, violence, and functional difficulties

Table 1 summarises participant characteristics for the overall sample and by dichotomised Wave 1 SDQ score. Male adolescents represented 47.7% of respondents. Most individuals were aged 11–14 years at Wave 1 (79.8%), and most had had two or three meals the previous day (81.8%). Most households owned some or all electric goods included in the survey's asset index, as well as some or all key household items (86.1%). 20.6% of adolescents had at least some difficulty with sight, hearing, mobility, communication, self-care, or cognition. At Wave 2, 8.7% of adolescents were categorised as depressed by the PHQ-A, and 84.5% reported experiencing at least one form of violence in the past year.

33.3% of adolescents in the top quintile of mental health difficulties at Wave 1 (SDQ top quintile cutoff = 22) experienced mental health difficulties at Wave 2, compared to 16.7% of other adolescents. 13.5% of those in the top quintile of mental health difficulties at Wave 1 screened positively for depression (PHQ-A score > = 10) compared to 7.6% of other adolescents, and 83.6% of those in the top quintile experienced past year violence at Wave 2, compared to 83.6% of other adolescents.

### What is the impact of Wave 1 poor mental health on Wave 2 outcomes?

Table 2 provides estimates of the impact of Wave 1 SDQ score on Wave 2 outcomes, where a higher Wave 1 SDQ score represents worse mental health. The risk difference and risk ratio represent the average change in risk expected for a one-unit increase in SDQ scores and illustrative absolute risks are provided for the median SDQ score. On average, we observed an increased risk of experiencing mental health difficulties, depression, and violence in at Wave 2 with higher SDQ-25 scores in early adolescence (Wave 1).

Across the sample, the average magnitude of this increased absolute risk of experiencing mental health difficulties in later adolescence was about 1.4%, while the relative risk increased on average by 7% (Table 2). For example, at the average baseline SDQ score of 17, the proportion of adolescents expected to experience mental health difficulties is 19.2% (Table 2). Were these individuals to have an SDQ score of 18, on average, the proportion expected to have mental health difficulties would be 20.6%, equivalent to a 7% increase in relative risk.

However, the actual magnitude of this increased actual risk depends on baseline SDQ score. Fig 1A–1C present the relationship between the SDQ score and the additional risk of experiencing each outcome. For example, examining Fig 1, we would expect 0.9% more young people to have mental health difficulties in later adolescence if their SDQ scores were 11 instead of 10 at Wave 1, while we would expect 2.3% more young people to have mental health difficulties in later adolescence if their SDQ scores were 31 instead of 30. The additional risk of having mental health difficulties in late adolescence is amplified for those already experiencing those difficulties in early adolescence.

We observed a similar pattern as described above for depression in later adolescence, with an average RD of 0.51% (0.30%, 0.72%) and RR of 1.06 (1.03, 1.08) (Fig 2).

However, we observed an inverse relationship with experiencing violence in the past year (Fig 3). The additional risk of experiencing violence still increases as SDQ scores increase, with an average RD of 0.49% (0.20%, 0.78%) and RR of 1.04 (1.01, 1.07). For this outcome, young

**Table 1. Description of CoVAC sample on exposure, outcomes, and covariates.**

| | | Total | Mental health difficulties (SDQ)–Wave 1 | |
|---|---|---|---|---|
| | | N (%) | Bottom 4 quintiles | Top quintile |
| Total | | 2773 | 2233 | 540 |
| **Covariates** | | | | |
| Gender–Wave 1 | | | | |
| | Male | 1322 (47.7) | 1064 (47.6) | 258 (47.8) |
| | Female | 1451 (52.3) | 1169 (52.4) | 282 (52.2) |
| Age group–Wave 1 | | | | |
| | 10 years or less | 161 (5.8) | 136 (6.1) | 25 (4.6) |
| | 11–14 years | 2214 (79.8) | 1780 (79.7) | 434 (80.4) |
| | 15 and above | 398 (14.4) | 317 (14.2) | 81 (15.0) |
| Violence intervention arm–Wave 1 | | | | |
| | No intervention | 1385 (49.9) | 1106 (49.5) | 279 (51.7) |
| | Intervention | 1388 (50.1) | 1127 (50.5) | 261 (48.3) |
| Meals on previous day–Wave 1 | | | | |
| | No meals | 11 (0.4) | 6 (0.3) | 5 (0.9) |
| | One meal or less | 415 (15.0) | 298 (13.3) | 117 (21.7) |
| | Two meals | 1092 (39.4) | 882 (39.5) | 210 (38.9) |
| | Three meals | 1176 (42.4) | 981 (43.9) | 195 (36.1) |
| | More than three meals | 76 (2.7) | 63 (2.8) | 13 (2.4) |
| | Missing | 3 (0.1) | 3 (0.1) | 0 (0.0) |
| Asset cluster–Wave 2 | | | | |
| | No household items & few electric goods | 28 (1.0) | 22 (1.0) | 6 (1.1) |
| | No electric goods but some household items | 357 (12.9) | 271 (12.1) | 86 (15.9) |
| | Some electric goods and some household items | 1878 (67.7) | 1534 (68.7) | 344 (63.7) |
| | High ownership of all items and good housing | 510 (18.4) | 406 (18.2) | 104 (19.3) |
| **Exposure** | | | | |
| Mental health difficulties (SDQ)–Wave 1 | | | | |
| | Lower difficulties | 2233 (80.5) | -- | -- |
| | Higher difficulties | 540 (19.5) | -- | -- |
| | Score: median (IQR) | 17 (14, 20) | 16 (13, 18) | 24 (23, 27) |
| **Effect modifier** | | | | |
| Functional difficulties (Washington Group)–Wave 1 | | | | |
| | No functional difficulties | 2203 (79.4) | 1821 (81.5) | 382 (70.7%) |
| | Any functional difficulties | 570 (20.6) | 412 (18.5) | 158 (29.3%) |
| **Outcomes** | | | | |
| Mental health difficulties (SDQ)–Wave 2 | | | | |
| | Lower difficulties | 2219 (80) | 1859 (83.3) | 360 (66.7) |
| | Higher difficulties | 554 (20) | 374 (16.7) | 180 (33.3) |
| | Score: median (IQR) | 8 (5, 12) | 8 (5, 11) | 11 (7, 15) |
| Depression score (PHQ-A)–Wave 2 | | | | |
| | Score < 10 | 2328 (84) | 1891 (84.7) | 437 (80.9) |
| | Score > 10 (indicative of depression) | 242 (8.7) | 169 (7.6) | 73 (13.5) |
| | Missing* | 203 (7.3) | 173 (7.7) | 30 (5.6) |
| Violence victimisation in the past year–Wave 2 | | | | |
| | No reported violence | 429 (15.5) | 367 (16.4) | 62 (11.5) |
| | Any reported violence | 2344 (84.5) | 1866 (83.6) | 478 (88.5) |

* A 'missing' row is shown only when missing data is present, e.g. in PHQ-A measures.

**Table 2. The expected average change in risk on the risk difference (RD) and risk ratio (RR) scales for an increase in SDQ score at Wave 1.** The leftmost column provides the observed proportion experiencing the given outcome at the median SDQ score (between 16–18 for most outcomes).

| Outcome | Proportion experiencing outcome at median W1 SDQ score | RD (95% CI) | RR (95% CI) | p-value for interaction term (Wald) |
|---|---|---|---|---|
| **Wave 2 SDQ score** | | | | |
| Overall sample | 19.2% | 1.41% (1.14%, 1.68%) | 1.07 (1.06, 1.09) | 0.11 |
| No Functional Difficulties at Wave 1 | 13.3% | 1.26% (0.96%, 1.57%) | 1.07 (1.06, 1.12) | -- |
| Functional Difficulties at Wave 1 | 18.4% | 1.93% (1.36%, 2.49%) | 1.09 (1.05, 1.08) | -- |
| **Wave 2 Depression** | | | | |
| Overall sample | 6.8% | 0.51% (0.30%, 0.72%) | 1.06 (1.03, 1.08) | 0.35 |
| **Wave 2 Past year violence** | | | | |
| Overall sample | 86.1% | 0.49% (0.20%, 0.78%) | 1.04 (1.01, 1.07) | 0.03 |
| No Functional Difficulties at Wave 1 | 82.7% | 0.34% (0.01%, 0.67%) | 1.00 (1.00, 1.01) | -- |
| Functional Difficulties at Wave 1 | 89.5% | 1.01% (0.40%, 1.62%) | 1.01 (1.00, 1.02) | -- |

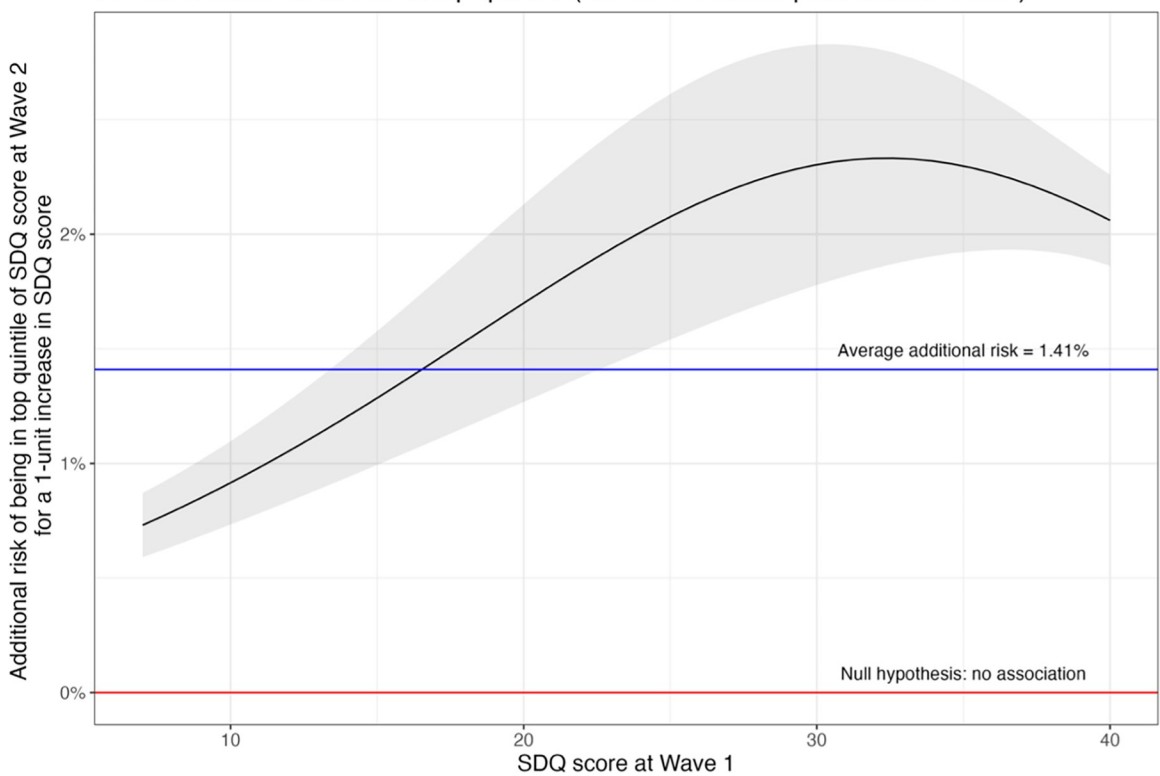

**Fig 1. The black line represents the additional absolute risk of being in the top quintile of poor mental health in later adolescence (y-axis) across all values of SDQ score in early adolescence after confounder adjustment, where higher SDQ score (x-axis) represents worse mental health in early adolescence.** The shaded area indicates 95% confidence intervals for the absolute risk. The blue line represents the average increase in absolute risk across all SDQ scores and corresponds to Table 1. The red line indicates the expected additional risk were there no association between mental health difficulties in early and late adolescence, e.g. no additional risk.

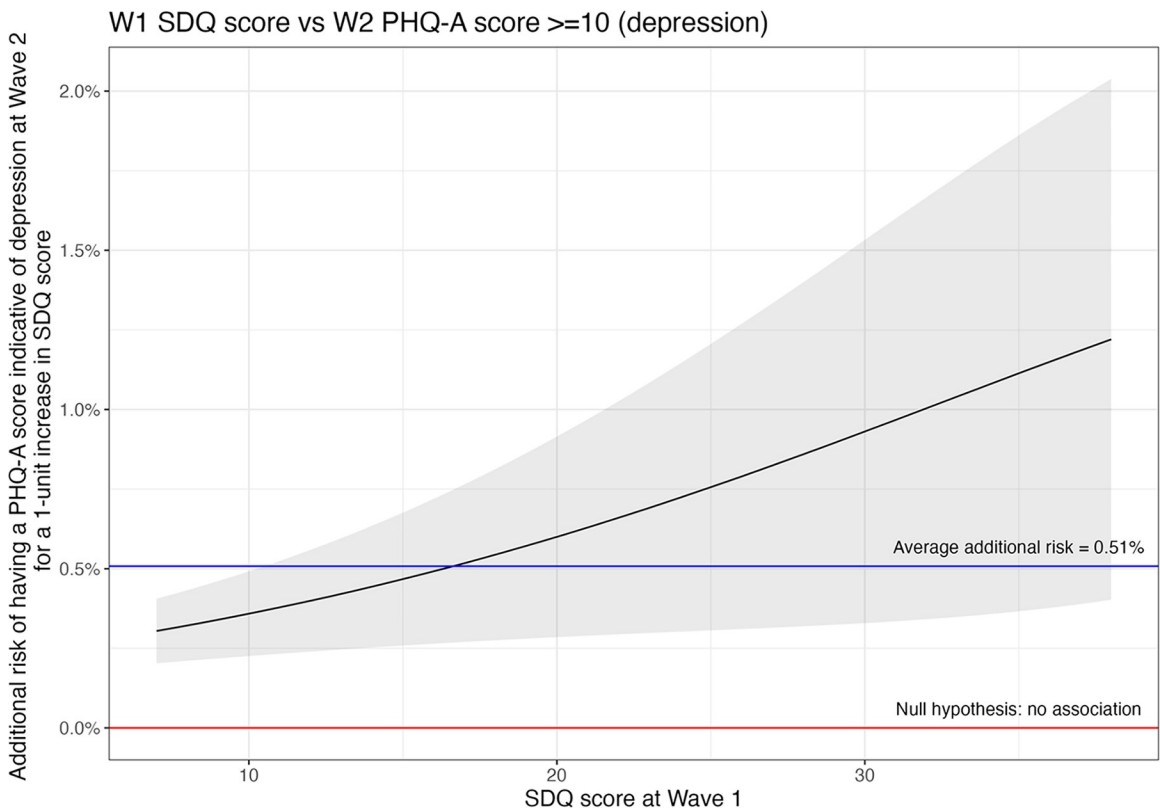

**Fig 2. The additional risk of having a PHQ score indicative of depression in later adolescence vs poor mental health in early adolescence where higher SDQ score (x-axis) represents worse mental health in early adolescence.**

people who experience fewer mental health difficulties in early adolescence have a greater additional risk of violence as their SDQ score increases.

## How does the relationship between Wave 1 poor mental health and Wave 2 outcomes differ by functional difficulty or disability?

There was some evidence of effect modification by functional difficulties, for the relationship between Wave 1 SDQ score and Wave 2 SDQ score (Table 2; Fig 4) and for the Wave 2 violence victimisation outcome (Table 2, Fig 5). There was no evidence of effect modification by functional difficulties for Wave 2 depression (Table 2) and therefore this figure is not shown.

For both outcomes where functional difficulties are an effect modifier, the average size of the impact of poor mental health at Wave 1 on the risk of poor mental health at Wave 2 and violence at Wave 2 was greater among those with functional difficulties at Wave 1 compared to those without. In particular, the risk profile for the violence outcome by SDQ score at Wave 1 was indicative of a linear increase in risk for young people without functional difficulties, but the additional risk of violence victimisation per unit increase in SDQ score was higher for young people with both functional difficulties and low SDQ scores (better mental health). This suggests that young people with functional difficulties with lower SDQ scores have greater additional risk of violence for small changes in their baseline mental health, while the risk is constant in young people without functional difficulties.

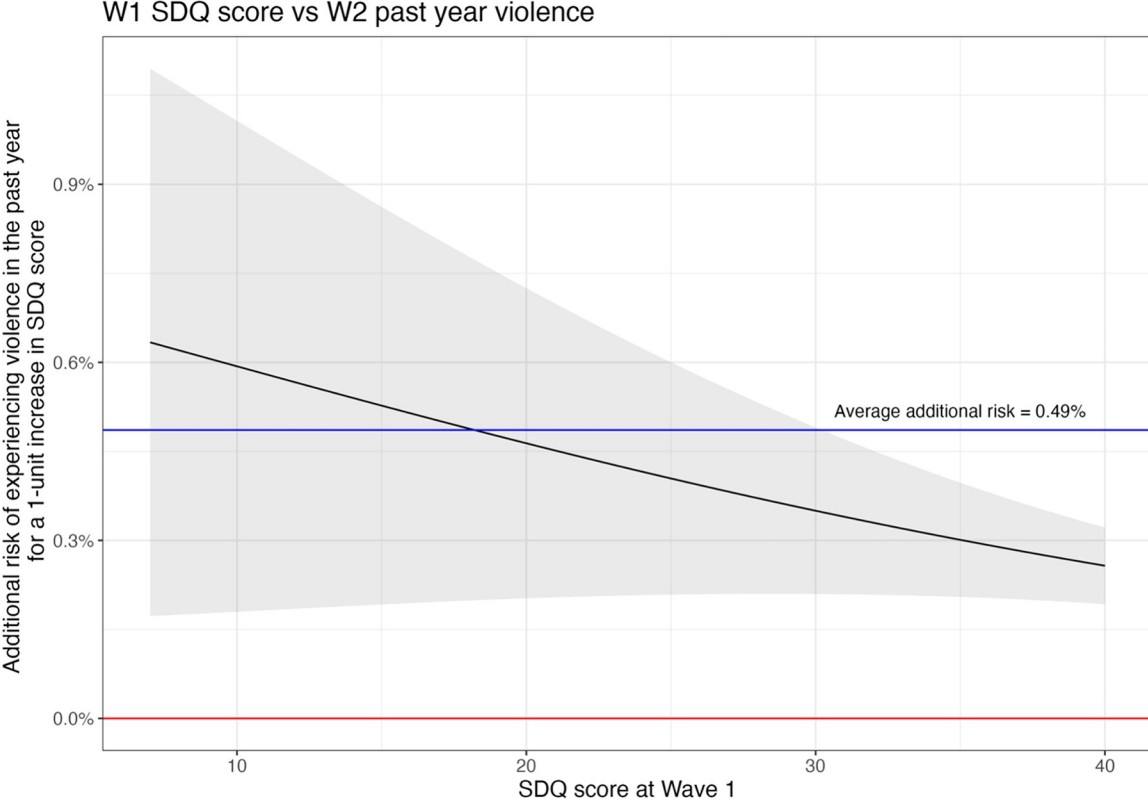

**Fig 3. The additional risk of past year violence victimisation in later adolescence vs poor mental health in early adolescence; NB: Higher SDQ score represents worse mental health.**

## Discussion

### Key findings

After controlling for key sociodemographic characteristics, we found an impact of mental health difficulties in early adolescence on risk of mental health difficulties, depression, and violence victimisation in later adolescence. Higher levels of mental health difficulties in early adolescence have a greater impact on the risk of mental health difficulties and depression outcomes than lower levels of mental health difficulties.

Conversely, higher levels of mental health difficulties in early adolescents have a lesser impact on the risk of past year violence compared to lower levels of mental health difficulties. We hypothesise that this may be attributable to a 'ceiling effect' whereby young people with higher mental health difficulties are already experiencing greater levels of violence and their risk therefore cannot increase further. Given high violence prevalence in this setting, it is likely that young people with high mental health difficulties have already experienced violence.

We also found evidence of effect modification by functional difficulties: poor mental health in early life had a greater impact on mental health difficulties and recent violence victimisation in later adolescence for young people with functional difficulties compared to young people with no functional difficulties.

### Comparison to other literature

To our knowledge, this is the first study of cohort data from outside a high-income country which has explored associations between early adolescent mental health and subsequent poor

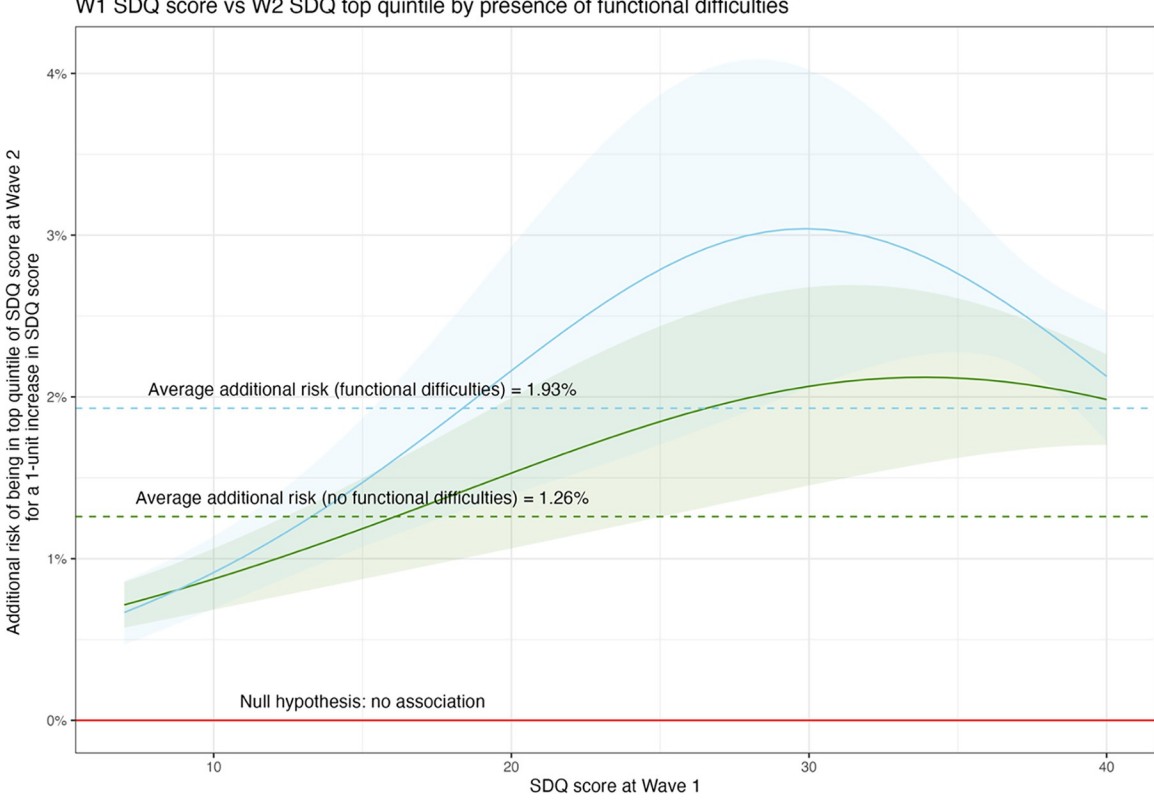

**Fig 4. The additional absolute risk of being in the top quintile of poor mental health in later adolescence (y-axis) across all values of SDQ score in early adolescence, where higher SDQ score (x-axis) represents worse mental health in early adolescence.** The shaded area indicates 95% confidence intervals for the absolute risk. The blue line represents the average increase in absolute risk across all SDQ scores for adolescents with functional difficulties and the green line for those without functional difficulties and corresponds to Table 1. The red line indicates the expected additional risk were there no association between mental health difficulties in early and late adolescence, e.g. no additional risk.

mental health and violence outcomes. Our findings are consistent in direction with previous cohort studies in the Netherlands, the UK and Finland, all of which find associations between adolescent mental health status and increased risk of later violence [38]. These have led to recommendations for clinical guidance for screening for violence experience and risk among adolescents with depression, and our results suggest that this may be appropriate for the Ugandan context as well.

Importantly, this is the first analysis of cohort data on effect modification by disability for associations between poor mental health and violence victimisation. Our findings suggest that adolescents with a disability are at a greater risk of persistent adverse mental health outcomes and increased violence victimisation over time. Previous research in high-income settings has indicated that having a disability in adolescence is associated with poorer mental health and global estimates suggest that children and adolescents with disabilities are between 2–4 times more likely to experience violence [12, 39]. Our work shows that disability is an extremely important factor to consider for interventions, policy and research for prevention of violence victimisation among those with poor mental health. The risks we observe suggest that adolescents with functional difficulties may remain on trajectories of poor mental health and have less resource/support available to modify these trajectories over time, including higher overall health expenses, economic poverty, and access barriers, relative to counterparts without

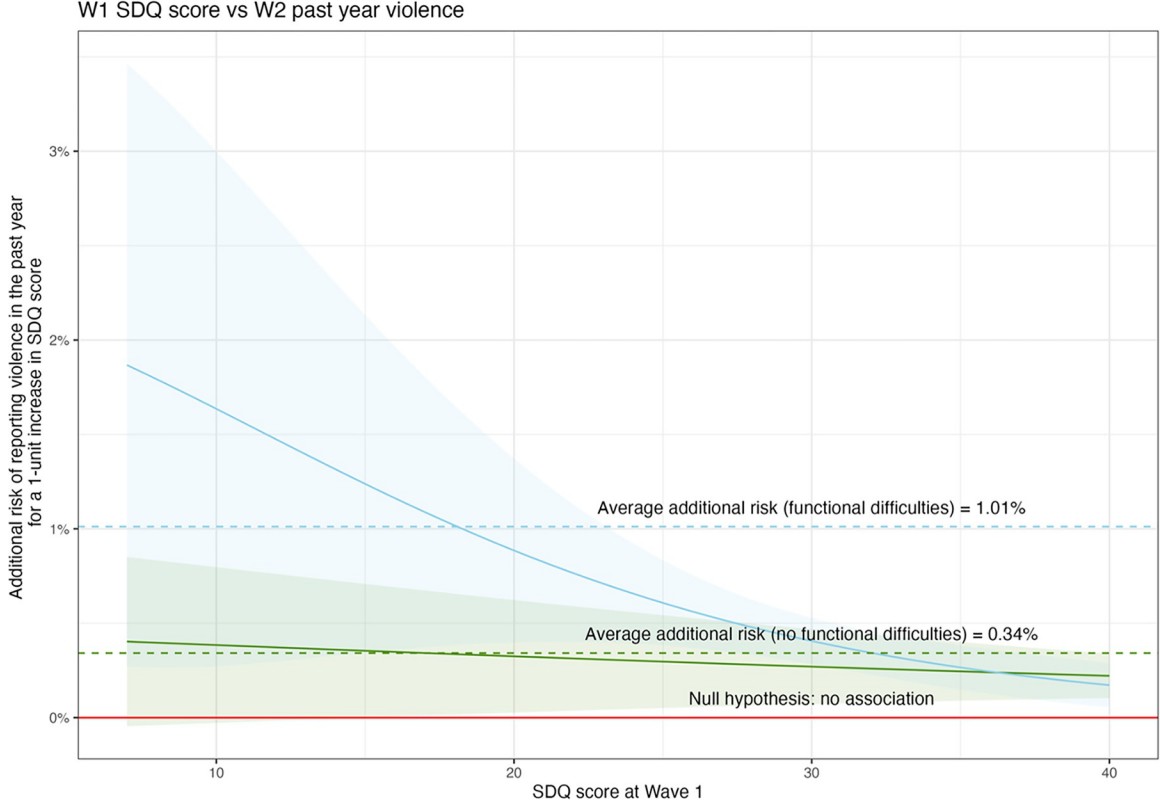

**Fig 5. The additional risk of past year violence victimisation in later adolescence vs poor mental health in early adolescence by functional difficulty or disability.**

functional difficulties [9, 40, 41]. Commitments to achieving international health targets, such as the WHO's Universal Healthcare Plan and the Sustainable Development Goals, cannot be achieved without reducing inequities between people with and without disabilities.

## Strengths and limitations

We make use of a large sample with prospectively collected data. Given the age and mobility of the sample, our attrition rate of 19.2% between Wave 1 and 2 is low. There were no clear differences in median SDQ score or in levels of violence victimisation in early adolescence between those who were included in the Wave 2 and those who were not.

We used validated measures of mental health difficulties, violence, and functional difficulties, but it is likely that there was still some under-reporting of these experiences by participants, especially where conditions are stigmatised. Our analysis made use of several sociodemographic covariates to adjust for confounding by age, sex, and socio-economic position. However, given that Wave 1 surveys were often completed by young adolescents, information pertaining to attributes they would not necessarily have much knowledge of in this context, such as household income or specific parental education, were not recorded. There is potential that we either over- or under-estimate the strength of association observed between mental health difficulties in early adolescence and subsequent mental health problems due to residual confounding in our analysis.

In this paper we used longitudinal data to test one model of directionality by limiting the temporality of the investigation to where onset of mental health symptoms clearly precedes

violence victimisation, however, there is evidence of a bidirectional association between violence victimisation and poor mental health that we have not explored in this study [5].

Our sample is drawn from Luwero District, which has a mixed urban/rural composition and demographic similarity to Uganda as a whole, compared to previous cross-sectional research in urban-dwelling adolescents [42]. However, Wave 1 participants were sampled from those attending primary 5, 6 or 7 in Luwero in 2014 and results should not be interpreted as generalisable to children not attending primary school [22].

All scales used in this research have been tested and validated, with many validated in the Ugandan context, but the potential remains for measurement error as with all studies that utilise psychometric scales.

While analysis of effect modification permitted identification of adolescents with disabilities as a potential target group for future mental health interventions, this study only investigated effect modification by Wave 1 disability and functional difficulties. This does not provide a complete picture of a person's ability as they move through the lifecourse, so disability at Wave 1 may not be commensurate with disability at Wave 2. Additionally, disability was measured with a questionnaire around functioning and some models of disability may place more importance on an adapted environment or social identity, particularly as it concerns mental health. While measured with a validated subscale, there may be some potential measurement error inherent to capturing the complex construct of disability as an effect modifier. Future research would benefit from analysis of other effect modifiers; for example, gender or ethnic group, both of which have are associated with inequalities in mental health outcomes in LMIC settings [43].

## Implications

Finally, there is significant scope for future research into the mechanisms governing the longitudinal relationship between poor mental health status and subsequent adverse outcomes, both in Uganda and globally. Future longitudinal research could investigate the role of mediating protective factors or contexts in the relationship between mental health issues in early and later adolescence. Identification of such mediators would further our understanding of the mechanisms through which poor mental health persists throughout adolescence and provide further valuable targets for mental health interventions and for interventions to improve the lives of young people living with disabilities. Qualitative and mixed-method approaches exploring the relationship between mental health, disability, and violence would also be valuable towards understanding young people's experiences of these intersecting areas, the barriers they face to seeking support, and which factors might be protective for early intervention.

In Uganda, where outpatient community mental healthcare options are limited, our findings highlight the importance of youth-centred mental health services and the potential for school-based interventions which can leverage existing infrastructure and relationships with school-aged young people, particularly vulnerable groups–like adolescents with disabilities–that may not have easy access to other healthcare settings [44, 45]. Importantly, health systems level efforts to improve access to health and mental health services should be responsive to the needs of young people living with disabilities. This is especially important, as young people living with disabilities can face financial, physical, communicative and sensory barriers to seeking healthcare support and services, and have higher associated health expenses [14, 41, 46].

The three-way relationship we observed between mental health, functional difficulties, and violence could be used to shape educational policy. For example, King and colleagues' 2018 analysis of a large Australian longitudinal cohort study found that bullying victimisation at school explained a large share of the relationship between disability status and poor mental

health in adolescents [47]. There is thus likely considerable potential for research into the role that collaboration between educational institutions and local health services can play in improving mental health programming in Uganda. Interventions such as the Good Schools Toolkit and others have employed this strategy to deliver both health and non-health interventions [35, 48]. Given the role that experiencing emotional or physical violence could play in placing adolescents with disabilities with mental health problems at greater future risk of mental health issues, these interventions to improve the mental health of disabled adolescents could use e.g. anti-bullying or anti-discrimination components in schools, or other approaches to violence prevention to reduce instances of violence victimisation involving both staff and students. These could include increasing awareness of disability in schools and interventions that focus on the specific needs in school environments of young people living with disabilities. Future research should consider the application of such interventions in the Ugandan context.

## Conclusions

This analysis has identified an important longitudinal relationship between mental health in early adolescence and mental health and violence in later adolescence in young people living in Uganda, particularly among those living with disabilities or other functional difficulties. These findings underscore the need for further research into the mechanisms through which mental health problems persist in vulnerable young people. They also highlight the potential and promise of preventing mental illness in childhood and adolescence as well as for the tailoring of mental health interventions to target these groups in particular. Such efforts will play a crucial role in improving the wellbeing of Ugandan youth during this important period in the lifecourse.

## Author Contributions

**Conceptualization:** Daniel J. Carter, Charlie F. M. Pitcairn, Jenny Parkes, Karen Devries.

**Data curation:** Louise Knight.

**Formal analysis:** Daniel J. Carter, Charlie F. M. Pitcairn.

**Funding acquisition:** Dipak Naker, Jenny Parkes, Karen Devries.

**Methodology:** Daniel J. Carter.

**Visualization:** Daniel J. Carter.

**Writing – original draft:** Daniel J. Carter, Charlie F. M. Pitcairn, Emily Eldred, Amiya Bhatia, Karen Devries.

**Writing – review & editing:** Daniel J. Carter, Charlie F. M. Pitcairn, Emily Eldred, Louise Knight, Janet Nakuti, Angel Mirembe, Lydia Atuhaire, Elizabeth Allen, Amiya Bhatia, Dipak Naker, Jenny Parkes, Karen Devries.

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
