## [Decision Letter · Decision Letter 0]

3 Sep 2024

PGPH-D-24-00886

Does disability modify the association between poor mental health and violence victimisation over adolescence? Evidence from the CoVAC cohort study in Uganda

Dear Dr. Carter,

Thank you for submitting your manuscript to PLOS Global Public Health. After careful consideration, we feel that it has merit but does not fully meet PLOS Global Public Health’s publication criteria as it currently stands. Therefore, we invite you to submit a revised version of the manuscript that addresses the points raised during the review process.

Please you are implored to critically consider each and every comment provided by the reviewers and revise your manuscript accordingly. Very importantly, make sure to highlight any changes to the revised draft for further assessment.

We look forward to receiving your revised manuscript.

Kind regards,

Razak M Gyasi, PhD, PD

Academic Editor

Journal Requirements:

1. Your current Financial Disclosure states, “Medical Research Foundation”. However, your funding information on the submission form indicates that you received funding from “Medical Research Council”. Please indicate by return email the full and correct funding information for your study and confirm the order in which funding contributions should appear. Please be sure to indicate whether the funders played any role in the study design, data collection and analysis, decision to publish, or preparation of the manuscript.

2. Please provide separate figure files in .tif or .eps format.

Additional Editor Comments (if provided):

Reviewers' comments:

Reviewer's Responses to Questions

**Comments to the Author**

1. Does this manuscript meet PLOS Global Public Health’s publication criteria? Is the manuscript technically sound, and do the data support the conclusions? The manuscript must describe methodologically and ethically rigorous research with conclusions that are appropriately drawn based on the data presented.

Reviewer #1: Yes

Reviewer #2: Yes

2. Has the statistical analysis been performed appropriately and rigorously?

Reviewer #1: Yes

Reviewer #2: Yes

3. Have the authors made all data underlying the findings in their manuscript fully available (please refer to the Data Availability Statement at the start of the manuscript PDF file)?

Reviewer #1: Yes

Reviewer #2: Yes

4. Is the manuscript presented in an intelligible fashion and written in standard English?

Reviewer #1: Yes

Reviewer #2: Yes

5. Review Comments to the Author

Reviewer #1: This is a very interesting research topic that has been rigorously carried out with high quality.

However, there is need to use politically correct terms throughout the manuscript before it is accepted for publication. The authors partially used this only in the case of people living with HIV but failed to use this politically correct term in the case of people living with disabilities. Please change all terms indicating people with disabilities to people living with disabilities.

Please state or provide the reliability or internal consistency for each research tool (SDQ PHQ-A, e Prevention of Child Abuse and Neglect-Child 150 Abuse Screening Tool etc) used in collecting the data.

Reviewer #2: this research was comprehensible and created more awareness on the importance on managing the mental wellbeing of Adolescence. Also, a gap in knowledge was identified and a research was effectively conducted which filled the gap and enlightened people on the need. The methodology and time use to conduct the research reflected the zeal to meet this need and attitude to do so. Finally, the findings of this study underscore the need for further research into the mechanisms through which mental health problems persist in young people and the need for people to engage in early intervention to prevent further damage and recurrent in later years.

6. PLOS authors have the option to publish the peer review history of their article (what does this mean?). If published, this will include your full peer review and any attached files.

**Do you want your identity to be public for this peer review?** For information about this choice, including consent withdrawal, please see our Privacy Policy.

Reviewer #1: **Yes: **Adeniyi Abolaji Adeboye

Reviewer #2: **Yes: **Blessing Jesubunmi Adeyemo

---

## [Decision Letter · Decision Letter 1]

26 Sep 2024

Does disability modify the association between poor mental health and violence victimisation over adolescence? Evidence from the CoVAC cohort study in Uganda

PGPH-D-24-00886R1

Dear Mr. Carter,

We are pleased to inform you that your manuscript 'Does disability modify the association between poor mental health and violence victimisation over adolescence? Evidence from the CoVAC cohort study in Uganda' has been provisionally accepted for publication in PLOS Global Public Health.

Best regards,

Professor Razak M Gyasi, PhD, PD

Academic Editor

Reviewer Comments (if any, and for reference):

Reviewer's Responses to Questions

**Comments to the Author**

1. If the authors have adequately addressed your comments raised in a previous round of review and you feel that this manuscript is now acceptable for publication, you may indicate that here to bypass the “Comments to the Author” section, enter your conflict of interest statement in the “Confidential to Editor” section, and submit your "Accept" recommendation.

Reviewer #1: All comments have been addressed

2. Does this manuscript meet PLOS Global Public Health’s publication criteria? Is the manuscript technically sound, and do the data support the conclusions? The manuscript must describe methodologically and ethically rigorous research with conclusions that are appropriately drawn based on the data presented.

Reviewer #1: Yes

3. Has the statistical analysis been performed appropriately and rigorously?

Reviewer #1: Yes

4. Have the authors made all data underlying the findings in their manuscript fully available (please refer to the Data Availability Statement at the start of the manuscript PDF file)?

Reviewer #1: Yes

5. Is the manuscript presented in an intelligible fashion and written in standard English?

Reviewer #1: (No Response)

6. Review Comments to the Author

Reviewer #1: All raised comments have been addressed

7. PLOS authors have the option to publish the peer review history of their article (what does this mean?). If published, this will include your full peer review and any attached files.

**Do you want your identity to be public for this peer review?** For information about this choice, including consent withdrawal, please see our Privacy Policy.

Reviewer #1: **Yes: **Adeniyi Abolaji Adeboye
